

# *Ooctonus vulgatus* (Hymenoptera, Mymaridae), a potential biocontrol agent to reduce populations of *Philaenus spumarius* (Hemiptera, Aphrophoridae) the main vector of *Xylella fastidiosa* in Europe

Xavier Mesmin[1,2], Marguerite Chartois[2], Guénaëlle Genson[2], Jean-Pierre Rossi[2], Astrid Cruaud[2] and Jean-Yves Rasplus[2]

[1] AGAP, INRAE, CIRAD, Montpellier SupAgro, Univ Montpellier, San Giuliano, France
[2] CBGP, INRAE, CIRAD, IRD, Montpellier SupAgro, Univ Montpellier, Montpellier, France

Corresponding author
Jean-Yves Rasplus,
jean-yves.rasplus@inrae.fr

## ABSTRACT

As a vector of *Xylella fastidiosa* (Wells, 1987) in Europe, the meadow spittlebug *Philaenus spumarius* (Linnaeus, 1758) (Hemiptera, Aphrophoridae) is a species of major concern. Therefore, tools and agents to control this ubiquitous insect that develops and feeds on hundreds of plant species are wanted. We conducted a field survey of *P. spumarius* eggs in Corsica and provide a first report of *Ooctonus vulgatus* Haliday, 1833 (Hymenoptera, Mymaridae) as a potential biocontrol agent of *P. spumarius* in Europe. To allow species identification, we summarized the main characters distinguishing *O. vulgatus* from other European species of *Ooctonus* and generated *COI* DNA barcodes. Parasitism rates were variable in the four localities included in the survey but could reach 69% (for an average number of eggs that hatched per locality of 109). Based on the geographic occurrences of *O. vulgatus* obtained from the literature, we calibrated an ecological niche model to assess its potential distribution in the Holarctic. Obviously, several questions need to be addressed to determine whether *O. vulgatus* could become an effective biocontrol agent of *P. spumarius* in Europe. So far, *O. vulgatus* has been reared only from *P. spumarius* eggs, but its exact host-range should be evaluated to ensure efficiency and avoid non-target effect. The top-down impact of the parasitoid on vector populations should also be assessed on large data sets. Finally, the feasibility of mass rearing should be tested. We hope this report serves as a starting point to initiate research on this parasitoid wasp to assess whether it could contribute to reduce the spread and impact of *X. fastidiosa* in Europe.

## INTRODUCTION

*Xylella fastidiosa* (Wells, 1987) is a xylem-dwelling insect-borne bacterium that originates from the Americas, infects more than 500 species of plants (*EFSA, 2015*) and causes a variety of scorch-like diseases in many cultivated species (*Almeida & Nunney, 2015*; *EFSA, 2018*; *Sicard et al., 2018*). Studies on the economic impact of *X. fastidiosa* have primarily focused on the wine and grape industries. Yield reduction and management costs to the California grape industry are estimated at more than US$100 million per year (*Tumber, Alston & Fuller, 2014*) and a potential introduction of the bacterium in Australia is estimated to cost up to AUD 7.9 billion over 50 years (*Hafi et al., 2017*).

X. fastidiosa has been recently detected in Europe and is present in Italy (*Saponari et al., 2013*), France (*Denancé et al., 2017*), Spain (*Olmo et al., 2017*), and Portugal (*DGAV, 2019*). Furthermore, niche modelling has shown that a large part of Europe is climatically suitable for the bacterium (*Godefroid et al., 2018*; *Godefroid et al., 2019*). Hence, *X. fastidiosa* represents a serious threat to European agriculture and natural ecosystems.

The spread of *X. fastidiosa* depends on several interacting factors, mainly insect vectors and plant communities as well as landscape, climate features and population dynamics of the bacterium itself (*Krugner et al., 2019*). As a consequence, disease management is complex. Reducing bacterium spread requires acting on a set of different biotic and abiotic factors (*Almeida et al., 2005*) and modelling approaches may help setting up effective strategies (*Fierro, Liccardo & Porcelli, 2019*). Here we focus on a possible management strategy to control populations of the most common vector of *X. fastidiosa* reported in Europe so far: the meadow spittlebug *Philaenus spumarius* (Linnaeus, 1758) (Hemiptera, Aphrophoridae) (*Saponari et al., 2014*; *Cornara et al., 2016*).

*P. spumarius* is highly polyphagous (*Cornara, Bosco & Fereres, 2018*), widely distributed in the Palearctic from sea level to high elevation (about 2,000 m; e.g., *Halkka, Raatikainen & Vilbaste, 1975*; *Lees, Dent & Gait, 1983*; *Drosopoulos & Asche, 1991*; *Loukas & Drosopoulos, 1992*; *Quartau, Borges & André, 1992*; *Stewart & Lees, 1996*; *Drosopoulos & Remane, 2000*), and was probably introduced to the New World (*Whittaker, 1973*). Its ability to acquire and transmit *X. fastidiosa* was previously demonstrated (*Severin, 1950*; *Saponari et al., 2014*; *Cornara et al., 2016*).

So far, a few studies have assessed the impact of different insecticides to reduce juvenile populations of *P. spumarius* in Europe (*Dongiovanni et al., 2018*; *Dader et al., 2019*). However, there is a growing awareness of the need to encourage management practices that safeguard harvests, human health, biodiversity and the environment. Thus, the development of effective biological control programs is desirable. Among biocontrol strategies, augmentative biological control consists in enhancing the effectiveness of naturally occurring natural enemies by the periodic release of specimens (*Eilenberg, Hajek & Lomer, 2001*; *Aubertot & Savary, 2005*). Compared to classical biological control it eliminates unintended effects of the introduction of new, non-native, parasitoids or predaceous arthropods (*Hoy, 2008*). However, as for all biological control programs, augmentative biocontrol requires field investigations to identify potential natural enemies of the target pest.

Currently, information about the natural enemies of the meadow spittlebug are scattered (*Cornara, Bosco & Fereres, 2018*). Species of birds, frogs, arachnids, and insects (Hymenoptera, Diptera, and Coleoptera) occasionally feed on *P. spumarius* (*Phillipson, 1960*; *Halkka & Kohila, 1976*; *Harper & Whittaker, 1976*; *Henderson, Hoffman & Jeanne, 1990*; *Pagliano & Alma, 1997*) but predation by native natural enemies does not appear to be an important source of mortality. Studies are in progress to test whether the invasive assassin bug *Zelus renardii* Kolenati, 1857 (Hemiptera, Reduviidae) could be used to control populations of *P. spumarius* in olive orchards (*Salerno et al., 2017*). However, mass release of this species may be risky for local biodiversity, especially for beneficial arthropods (*Ables, 1978*). Indeed, it is considered as a generalist predator (*Ables, 1978*; *Cisneros & Rosenheim, 1998*; *Weirauch, Alvarez & Zhang, 2012*; *Salerno et al., 2017*, but see *Cohen & Tang, 1997* who suggest a strong effect of prey body size).

So far, only few parasitoids of *P. spumarius* have been recorded. Adults are attacked by *Verralia aucta* (Fallen, 1817) (Diptera, Pipunculidae) in Europe with relatively high parasitism rates in England: in average 31% in females and 46% in males over four years (*Whittaker, 1969*; *Whittaker, 1973*). Parasitism by *V. aucta* has a direct effect on *P. spumarius* population dynamics because it renders the host sterile (*Whittaker, 1973*). However, this parasitoid does not have an immediate effect on bacterium transmission because adults are only killed after 10–11 weeks of parasitism (*Whittaker, 1969*), a period during which they are probably still able to spread the bacterium. Contrastingly, an interesting feature of egg parasitoids is that they kill the host in the egg stage, that is, before it can inflict damage to its host plants (*Mills, 2010*). In the case of *P. spumarius*, the insect is killed before it acquires the bacterium from an infected host plant and becomes able to transmit it. A few egg parasitoids have been recorded in the US: *Ooctonus vulgatus* Haliday, 1833 (Hymenoptera, Mymaridae) and at least two unnamed species of *Centrodora* (Hymenoptera, Aphelinidae) (*Weaver & King, 1954*). Indeed, the genus *Tumidiscapus*, which is cited as parasitoid of *P. spumarius* in the US (*Weaver & King, 1954*), is in fact a synonym of *Centrodora* (*Hayat, 1983*). However, little is known about the biology and efficacy of egg parasitoids *in natura*.

In this study, a field survey was conducted to identify major egg parasitoids of *P. spumarius* in Corsica. We provide a first report of *Ooctonus vulgatus* in this area. We summarized the main characters separating *O. vulgatus* from other Palearctic species to facilitate identification and generate *COI* DNA barcodes to accurately identify the species. Finally, we reviewed the literature and gathered all available occurrence data (i.e., geographical coordinates) of previously detected populations of *O. vulgatus*. This allowed us to calibrate an ecological niche model linking different climate descriptors to species occurrence data and estimate the potential distribution of the parasitoid in the Holarctic region for comparison with the distribution of *P. spumarius*.

## MATERIALS AND METHODS

### Sampling and calculation of parasitism rate

Five to ten handfuls of about eight top branches of *Cistus monspeliensis* L. 1753 (cut at 50 cm below the end of the branch) were sampled in four localities (Fig. 1). These localities

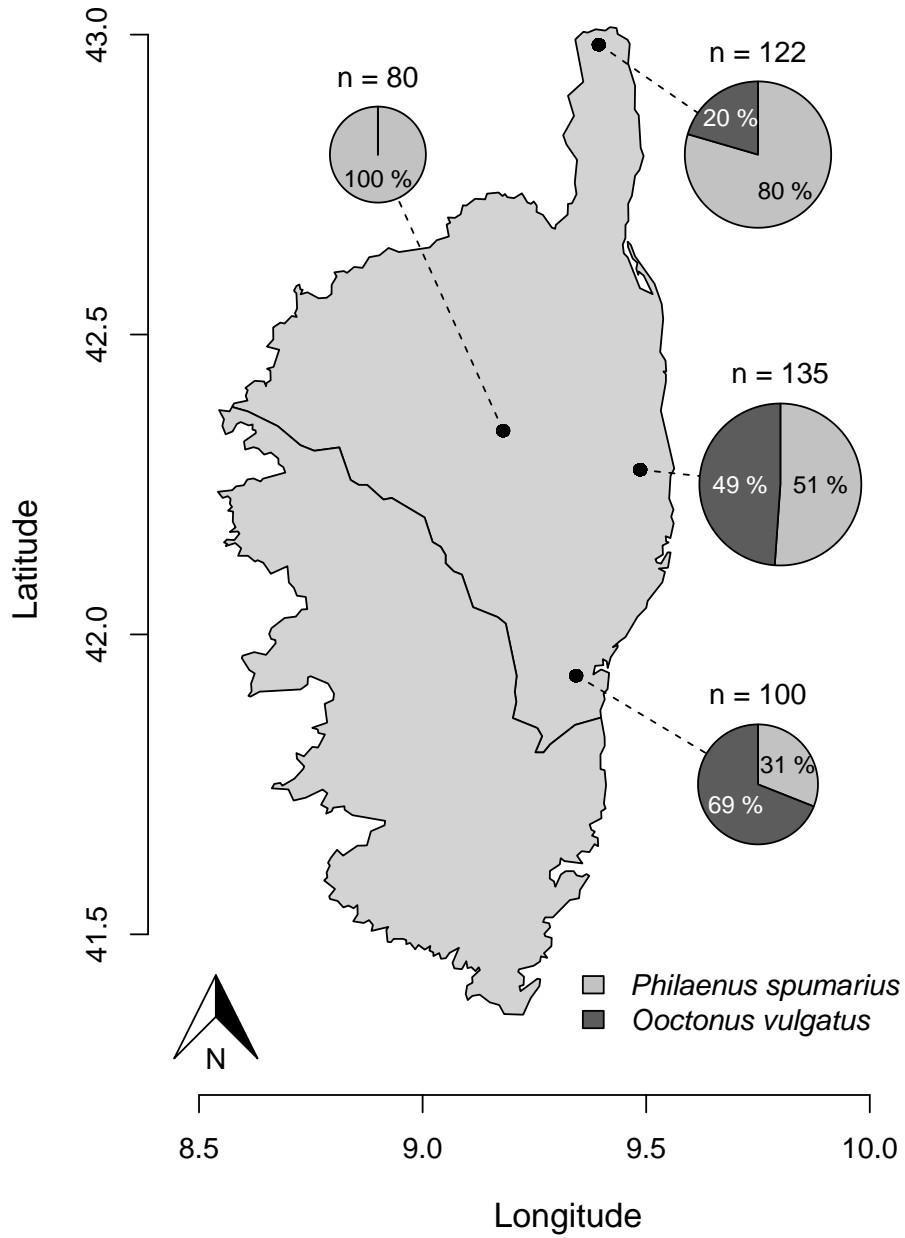

**Figure 1 Parasitism rate of *Philaenus spumarius* eggs in the four sites sampled in Corsica.** Size of the pie chart is proportional to the total number of eggs that hatched from each locality (n). Slices indicate the relative proportion of *O. vulgatus* (dark grey) and nymphs of *P. spumarius* (light grey) that emerged from the pool of eggs. GPS coordinates of sampling localities: 42.984205°N, 9.395287°E (Ersa); 42.338849°N, 9.180636°E (Tralonca); 42.274756°N, 9.487185°E (Canale-di-Verde); 41.931726°N, 9.343731°E (Ventiseri). The map was built with the R package maps, using data from UNESCO (1987) through UNEP/GRID-Geneva.

were part of a larger field survey of population dynamics of *P. spumarius* in Corsica. We targeted *C. monspeliensis* to maximize our chances to find eggs of *P. spumarius*. Indeed, we demonstrated in a previous study that, in Corsica, adults of *P. spumarius* seemed to be mainly associated with this species (*Cruaud et al., 2018*). Sampling was performed between the 12th and the 15th of February 2019. The back of each leaf (about 900 leaves per handful of branches) was inspected in the laboratory for whitish clusters, which were retained and inspected under a binocular microscope to confirm the presence of eggs of *P. spumarius* (Appendix S1; 109, 148, 167, 187 eggs obtained per site, hence a total of 611 eggs monitored). The morphological identification of *P. spumarius* eggs and first stage nymphs was performed using the descriptions of *Weaver & King (1954)* (Appendix S1). The pieces of leaf containing the eggs were placed on filter papers in unaerated Petri dishes (i.e., without spur) at room temperature ($20.2 \pm 1.5\,°C$), with natural light. Filter papers were kept moist by adding drops of water when necessary. Hatching was monitored every morning from the 18th of February to the 15th of March 2019. Emerging nymphs and parasitoids were killed and stored in 70% Ethanol at $4\,°C$. Parasitism rates were computed in each locality with the following formula: $Parasitism\ rate = \frac{Number\ of\ parasitized\ eggs}{Number\ of\ parasitized\ eggs + Number\ of\ unparasitized\ eggs}$ (*Costello & Altieri, 1995*).

## Morphological identification of the parasitoids

Identification to species was performed using the *Ooctonus* keys by *Triapitsyn (2010)* and *Huber (2012)*. Specimens were desiccated using HMDS (*Heraty & Hawks, 1998*) and glued on grey cards. Imaging was performed with a Keyence digital microscope (VHX-5000 Camera color CMOS and the VH-Z100UT lens). Images were then edited in Adobe Photoshop CS6©software.

## Molecular identification of the parasitoids

Six individuals were used for molecular identification. Three of them were handled individually (sample codes = XMES00042_0101, XMES00077_0101, XMES00091_0101) and the remaining three were pooled to increase DNA yield (sample code = XMES00041_0189). Total genomic DNA was isolated using the Qiagen DNeasy Blood & Tissue kit without destruction of the specimens. We followed manufacturer's protocol with the following modifications. Samples (whole insects, without dissection or crushing) were incubated overnight in an Eppendorf thermomixer (temperature = $56\,°C$, mixing frequency = 300 rpm). To increase DNA yield, two successive elutions ($50\,\mu L$ each) were performed with heated buffer AE ($56\,°C$) and an incubation step of 15 min followed by centrifugation (6,000 g for 1 min at room temperature; see *Cruaud et al. (2019)* for a detailed description of the protocol). Eppendorf microtubes LoBind 1.5 ml were used for elution and to store DNA at minus $20\,°C$ until PCR amplification. Vouchers were deposited at Centre de Biologie pour la Gestion des Populations (CBGP), Montferrier-sur-Lez, France. The mitochondrial Cytochrome c oxidase I standard barcode fragment (*COI*) was amplified with a cocktail of M13-tailed primers as detailed in *Germain et al. (2013)*. Unpurified PCR products were sent to Eurofins MWG Operon (Ebersberg, Germany) for sequencing using the M13F and M13R primers (*Germain et al., 2013*; *Ivanova et al.,*

*2007*). Both strands for each overlapping fragment were assembled in Geneious v11.1.4 (https://www.geneious.com). Geneious was also used to translate consensus sequences to amino acids to detect premature codon stops. All *COI* sequences available on BOLD (*Ratnasingham & Hebert, 2007*) for *Ooctonus* species were downloaded (last access July 12, 2019) and aligned with the newly generated sequences using MAFFT v7.245 (*Katoh & Standley, 2013*). A maximum likelihood tree was inferred with raxmlHPC-PTHREADS-AVX version 8.2.4 (*Stamatakis, 2014*). A rapid bootstrap search (100 replicates) followed by a thorough ML search (-m GTRGAMMA) was conducted. Tree visualization and annotation was performed with TreeGraph 2.13 (*Stöver & Müller, 2010*).

## Species distribution modelling framework

Occurrences of *O. vulgatus* were retrieved from the literature and the GBIF database (*GBIF.org, 2019*) (Tables S1 and S2). Two hundred and five occurrences were obtained from the literature (Table S2), eight of which were not included in the analysis as no geographic coordinates were available. Forty occurrences were obtained from GBIF (last access: 22 August 2019; Table S2), but were all discarded due to dubious identification or lack of information on sample origin. Therefore, no occurrence obtained from GBIF could be included in the analysis.

We fitted a correlative model linking different climate descriptors to species occurrences. The Maxent algorithm was chosen to conduct analyses because it does not require absence data (i.e., locations in which we can presume that a species is truly absent) (*Phillips, Anderson & Schapire, 2006*). We summarized below the main step of our analysis and details are provided in Appendix S2. The mean temperature and precipitation of the wettest, driest, warmest, and coldest quarters as well as precipitation seasonality were extracted from the Worldclim 2.0 database (*Fick & Hijmans, 2017*) and used as bioclimatic descriptors (*Hijmans et al., 2005*). In absence of formal knowledge about climatic factors constraining *O. vulgatus* distribution, we constituted three sets of bioclimatic variables and performed modelling with each of them (*Qiao, Soberón & Peterson, 2015*; *Godefroid et al., 2019*). The first set (CLIM1) comprised the mean temperature of the wettest, driest, warmest, and coldest quarters to reflect the impact of temperature constraints on distribution. To highlight the precipitation constraint, we added the precipitation seasonality to CLIM1 and constituted the second set (CLIM2). Finally, we built a third set (CLIM3) by assembling CLIM1 and the precipitation of the wettest, driest, warmest, and coldest quarters to fully account for both extreme temperatures and precipitations in the species distribution models (SDMs). The Maxent algorithm requires a set of locations where the species has been found (here, a random 70% of the available occurrences, the other 30% being used for model validation) and a set of locations where no information about the presence of the species are available (referred to as background points). A total of 10,000 background points were randomly generated in North America and Europe. To render complex response to environmental constraints while reducing model overfitting we first fitted 48 Maxent models using six regularization multiplier (RM) combinations (L, LQ, H, LQH, LQHP, LQHPT with L = linear, Q = quadratic, H = hinge, P = product and T = threshold) and feature class (FC) values (eight values ranging from 0.5 to 4 with

increments of 0.5) (*Radosavljevic & Anderson, 2014*). Optimal FC and RM combinations were determined for each of the three bioclimatic datasets (CLIM1–CLIM3) using the R language (*R Core Team, 2019*) and the package ENMeval (*Muscarella et al., 2014*). Optimal parameters were then used to fit a set of 10 replicate Maxent models using 70% of the dataset. The performance of each model was evaluated using the remaining 30% of occurrences using the area under the receiver–operator curve (AUC, *Fielding & Bell, 1997*) and the true skill statistics (TSS, *Allouche, Tsoar & Kadmon, 2006*). Models with AUC <0.8 were excluded from further analyses (*Vicente et al., 2013*). Habitat suitability maps (logistic output ranging from 0 to 1) were transformed into binary projections using the threshold that optimized the TSS statistics on the testing data (*Guisan, Thuiller & Zimmermann, 2017*). Maxent replicate models were fitted and evaluated using the R package biomod2 (*Thuiller et al., 2009*).

Two different outputs were generated using the set of model prediction. (i) Binary predictions were averaged to produce the committee (consensus) averaging (*Araújo & New, 2007*; *Marmion et al., 2009*) showing the likelihood of the presence of *O. vulgatus*. This consensus model ranges from 0 (all the models predict absence) to 100% (all the models predict presence) and (ii) the median of the logistic outputs (*Guisan, Thuiller & Zimmermann, 2017*) of the models that depicts the climate suitability across the different models.

# RESULTS

## Parasitism rates

Out of the 611 eggs monitored, 437 (i.e., 71.5%) hatched. 277 (63.4%) gave rise to *P. spumarius* nymphs and parasitoids emerged from 160 eggs (36.6%). All parasitoids were identified as *O. vulgatus* (Fig. 2). No parasitoid emerged from eggs collected in one of the four localities. We observed parasitism rates of 20.5, 48.9 and 69.0% in the three other localities (Fig. 1).

## Guidelines for the identification of *O. vulgatus*

To help identification, we list below the main features that differentiate *O. vulgatus* from its closest relatives. The genus *Ooctonus* has been recently revised in the Palearctic and Nearctic regions respectively by *Triapitsyn (2010)* and *Huber (2012)*. *Ooctonus* can be distinguished from other genera of Mymaridae by the following set of characters: tarsi 5-segmented, propodeum with diamond-shaped pattern of carinae (Fig. 2F), fore wing venation about one-third the wing length (Fig. 2C), with short marginal and stigmal vein, parastigma with hypochaeta next to proximal macrochaeta (*Huber, 2012*). In the Holarctic region, *O. vulgatus* can be distinguished from other species of *Ooctonus* by the following unique combination of features (Fig. 2): vertex without stemmaticum; mesoscutum without median groove; posterior part of scutellum and frenum smooth with weak sculpture laterally; metanotum and propodeum without reticulate sculpture; propodeum without median carina, but with a pentagonal areole formed by dorsolateral carinae; short petiole, 0.9–1.2x as long as metacoxa; forewing at least slightly truncate apically; females funicle with multiporous placoid sensilla (mps) on F7 and F8 only, F5 and F6 without mps; single

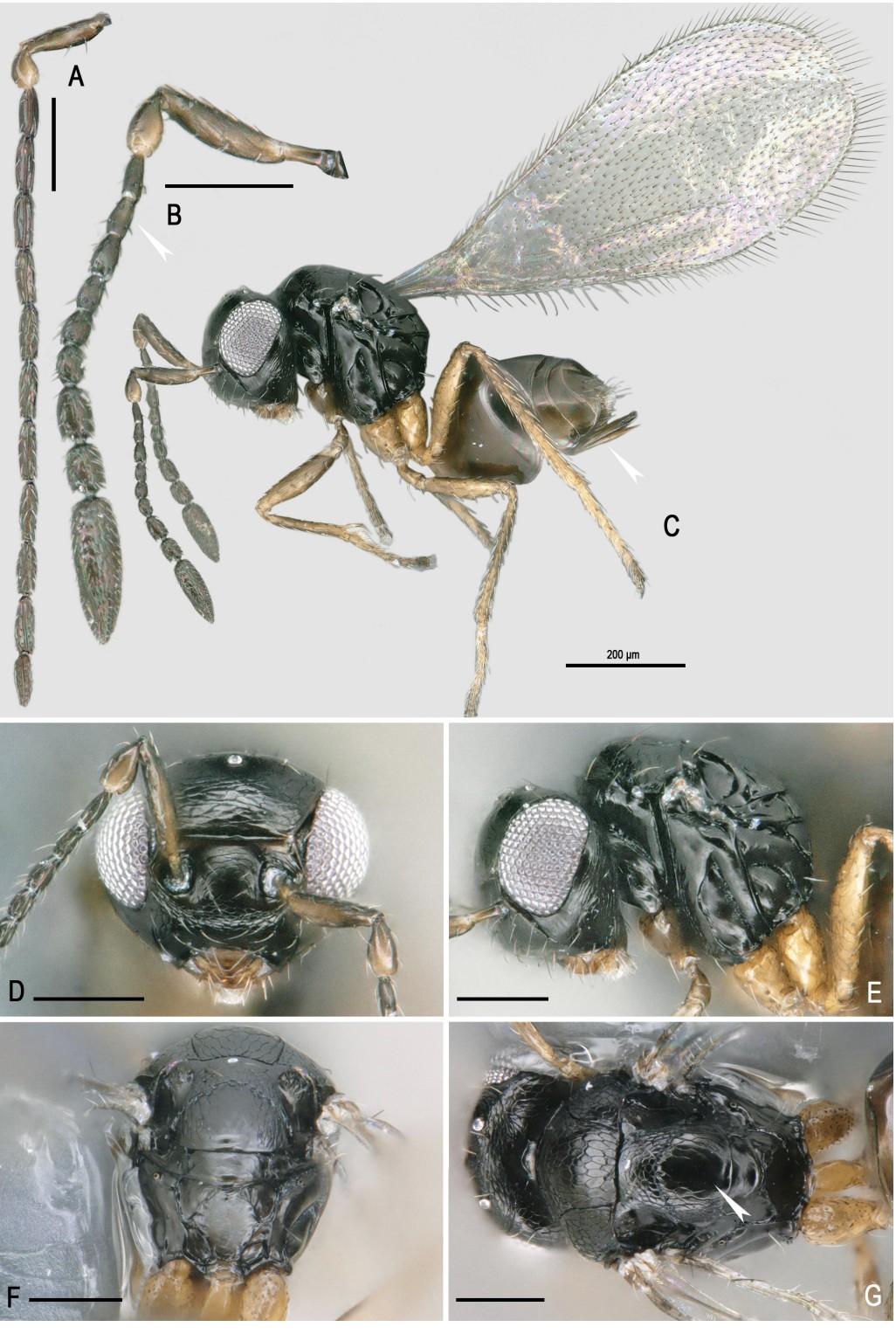

**Figure 2** **Morphology of *Ooctonus vulgatus* Haliday, 1833.** (A) Male antenna (B) Female antenna (C) Habitus. (D) Head front view. (E) Mesosoma lateral view. (F) Male propodeum. (G) Mesosoma dorsal view. All scales = 100 μm except habitus. Photo credit: Jean-Yves Rasplus INRA.

row of six bullae inside the female clava; ovipositor at most 1. 4× as long as metatibia and only slightly exerted beyond apex of gaster.

## Molecular identification of the parasitoid

Barcode sequences were successfully generated from all samples. All sequences were identical. Phylogenetic analysis confirmed that the most likely identification was *O. vulgatus* (Fig. S1).

## Species distribution modelling

A total of 200 occurrences (197 obtained from the literature plus the three localities where we sampled *O. vulgatus*) (Fig. 3A) were used to model the distribution of *O. vulgatus* in Europe. The optimal Maxent parameters were RM = 4 and FC = hinge; RM = 4 and FC = hinge and RM = 2.5 and FC = hinge for CLIM1, CLIM2, and CLIM3, respectively. With the exception of one model of CLIM2, all models based on these optimal values yielded AUC values >0.8, which indicated that the different bioclimatic data subsets performed well. The consensus model was therefore computed from a set of 29 estimates of climate suitability.

Figure 3B shows the median of the climate suitability values for the 29 models considered. Figure 3C depicts the proportion of the 29 models indicating that the climate is suitable for *O. vulgatus*. Both Figs. 3B and 3C show that the climate is favorable in very large areas covering most of Western Europe and around the Black Sea. These areas are overlapping with the geographical range of *P. spumarius* (*Cruaud et al., 2018*).

## DISCUSSION

*Ooctonus* Haliday, 1833 is a medium-sized genus of Mymaridae containing 37 described species that occur in all zoogeographical regions of the world (*Holt et al., 2013*) except Australasia (*Noyes, 2019*). *O. vulgatus* has been reared from the eggs of *P. spumarius* and studied only once in North America (*Weaver & King, 1954*). This species is thus poorly known as confirmed by the limited barcoding record. Indeed, only four barcodes are available in BOLD (two from Virginia United States, one from Ontario Canada, and one from British Columbia Canada). As a likely component of aerial plankton, *O. vulgatus* is expected to be a widespread species distributed in the Holarctic region (ranging from Ireland to the Sakhalin peninsula and from eastern to western coasts of North America, as south as California (*Huber, 2012*)). The species has been also reported from China (*Bai, Jin & Li, 2015*) but authors' illustration casts some doubts about specimen identification. There are only a few unquestionable occurrences in the literature for this species ($n = 197$). Here, we provide a first report of *O. vulgatus* in Corsica and assess, for the first time in Europe, its biology as parasitoid of *P. spumarius*. We also confirm its potential large distribution throughout Europe with modelling approaches. More importantly we show that *O. vulgatus* potential distribution in Europe (Fig. 3) overlaps that of its host *P. spumarius* (*Cruaud et al., 2018*), which is not surprising from a biological point of view but is an interesting result in the framework of biological control. This study is preliminary and predictions, especially because they are based on a limited number of occurrences, are indicative only. This study
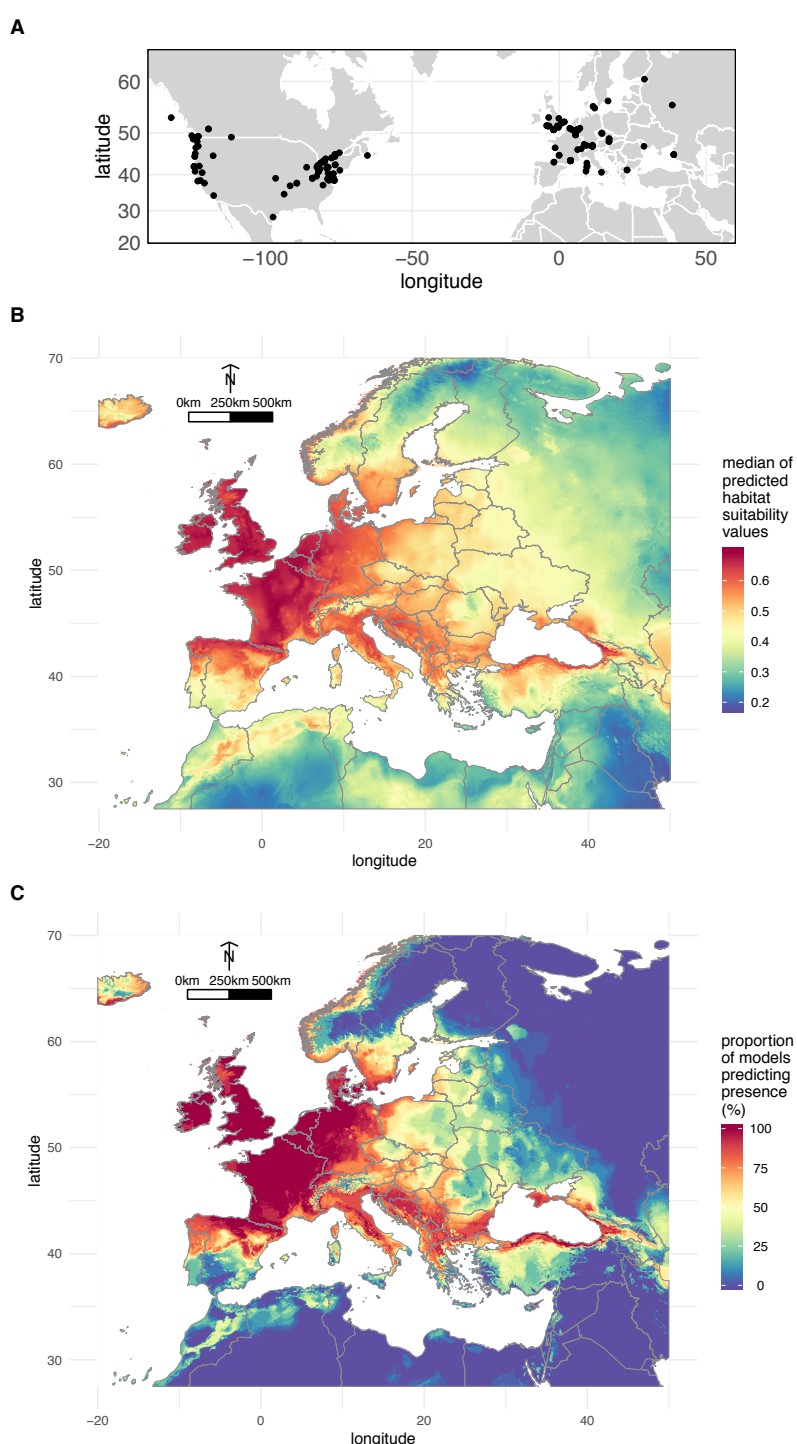

**Figure 3** **Geographical distribution of *O. vulgatus*.** (A) Distribution of *O. vulgatus* occurrences collected from the literature. (B) Consensus model of climate suitability estimated by Maxent: median of model outputs. (C) Consensus model of climate suitability estimated by Maxent: proportion of models predicting *O. vulgatus* presence in Europe.

is a starting point to encourage investigations in other parts of Europe. Sampling efforts should more specifically target areas predicted as suitable for *P. spumarius* but non-suitable for *O. vulgatus* such as eastern areas of Europe.

When studied in North America, observed parasitism rates did not exceed 10% of the sampled eggs of *P. spumarius* (*Weaver & King, 1954*). Here, we obtained parasitism rates of up to 69%, but absence of parasitism in one sampled site. While we acknowledge sampling four sites is not enough to have a representative view of *P. spumarius* egg parasitism in Corsica, our results show that parasitism rate can be high though very variable. Further surveys are obviously necessary to better assess the spatial and temporal variability of parasitism rate and understand what is(are) the cause(s) of such variations in Corsica and throughout Europe. Identifying such drivers could open new avenues for conservation biological control against *P. spumarius*, through the implementation of environments favorable to *O. vulgatus* in the vicinity of crops susceptible to *X. fastidiosa*.

The use of mymarids in biological control program has a long history. The most notable instance being the use of *Anaphes nitens* (Girault, 1928) in several countries to successfully control the eucalyptus weevil, *Gonipterus scutellatus* Gyllenhal, 1833 (Coleoptera, Curculionidae), which feeds and reproduces on *Eucalyptus* trees (*Doull, 1955*). More recently, *Cleruchoides noackae* Lin and Huber, 2007 has been used in South America to control an invasive sap-feeding pest of *Eucalyptus*, *Thaumastocoris peregrinus* Carpintero and Dellapé, 2006 (Hemiptera, Thaumastocoridae) (*Martinez, González & Dicke, 2018*). Mymarid species were used to control leafhopper vectors of plant pathogens (Hemiptera, Cicadellidae). *Anagrus armatus* (Ashmead, 1887) regulated *Edwardsiana froggatti* (Baker, 1925) (Hemiptera, Cicadellidae), a pest of apple in New Zealand, with parasitism rates of the eggs reaching 80% (*Dumbleton, 1937*). More recently, *Cosmocomoidea* species were used to target *Homalodisca vitripennis* (Germar, 1821) (Hemiptera, Cicadellidae) a vector of *X. fastidiosa* in California (*Irvin & Hoddle, 2010*). In all these cases, mymarids helped regulate pest population growth.

However, before any attempts to regulate populations of *P. spumarius* are made, we need to enrich our knowledge on *O. vulgatus*. In particular, the degree of specificity of the *P. spumarius* – *O. vulgatus* interaction needs to be determined to avoid non-target effect of augmentative biocontrol (*Van Driesche & Hoddle, 2016*). We also need to evaluate our ability to consistently rear *O. vulgatus* in controlled conditions, one of the key obstacles to the use of mymarids in biological control programs (but see *Martinez, González & Dicke, 2018*). Finally, parasitoids can have complex effects on vector-borne disease by either increasing (*Jeger et al., 2011*) or decreasing (*Martini, Pelz-Stelinski & Stelinski, 2014*) pathogen spread. Further research is still needed to better understand the impact of such tri-trophic interactions on plant disease dynamics. While *O. vulgatus* does not directly impact transmission capacity of *P. spumarius*, by killing its host at an early stage of development, it reduces the number of vectors that may acquire the bacterium from an infected host-plant and become able to transmit it.

Again, we consider this study as a starting point to encourage research on this parasitoid wasp to assess whether it could contribute to reduce the spread and impact of *X. fastidiosa* in Europe. Increasing egg parasitism of *P. spumarius* in the fall might significantly reduce

![PeerJ]

population size in the next year and possibly the transmission of the bacterium, without resorting to chemical treatments.

### Funding

This work was funded by the Collectivité Territoriale de Corse and the European Union Horizon 2020 research and innovation program under Grant Agreement No. 727987 XF-ACTORS. The funders had no role in study design, data collection and analysis, decision to publish, or preparation of the manuscript.

### Grant Disclosures

The following grant information was disclosed by the authors:
Collectivité Territoriale de Corse and the European Union Horizon 2020 research and innovation program:  727987 XF-ACTORS.

### Competing Interests

The authors declare there are no competing interests.

### Author Contributions

- Xavier Mesmin conceived and designed the experiments, performed the experiments, analyzed the data, prepared figures and/or tables, authored or reviewed drafts of the paper, and approved the final draft.
- Marguerite Chartois analyzed the data, prepared figures and/or tables, authored or reviewed drafts of the paper, and approved the final draft.
- Guénaëlle Genson performed the experiments, analyzed the data, authored or reviewed drafts of the paper, and approved the final draft.
- Jean-Pierre Rossi, Astrid Cruaud and Jean-Yves Rasplus conceived and designed the experiments, analyzed the data, prepared figures and/or tables, authored or reviewed drafts of the paper, and approved the final draft.

### Data Availability

    The COI sequences are available at NCBI: MN641903–MN641906.

### Supplemental Information

Supplemental information for this article can be found online at http://dx.doi.org/10.7717/peerj.8591#supplemental-information.

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
