# Peer review of "Ooctonus vulgatus (Hymenoptera, Mymaridae), a potential biocontrol agent to reduce populations of Philaenus spumarius (Hemiptera, Aphrophoridae) the main vector of Xylella fastidiosa in Europe"

_PeerJ, doi:10.7717/peerj.8591_

## Round 0.1 · original submission · Major Revisions

This is an interesting report on the emergence of the parasitoid wasp species, Ooctonus vulgatus, from eggs of the spittlebug Philaenus spumarius, vector of the plant pathogen Xylella fastidiosa.

As stated by the authors, this study represents a starting point for further investigations on the possibility for O. vulgatus to be used as a biocontrol agent of the Xylella vector, even if in some part of the text the parasitoid is clearly defined as a promising biocontrol agent. I believe that this statement is not fully supported by the data presented in the article and I would also suggest the authors to provide “a more careful” title e.g., "Ooctonus vulgatus, an egg parasitoid of Philaenus spumarius…".

In my opinion, some important information is also missing and materials and methods should be better clarified and provided with more details.
For example, it seems that DNA isolation has been carried out without samples desruction: this should be better clarified. More details are needed for species distribution modelling, too.

The authors should also consider that since they carried out a single survey of parasitoid emergence from of P. spumarius eggs, i.e. in absence of a quantitative approach, they shouldn’t refer to a “rate of parasitism”. In addition, it is worth mentioning that the use of the egg parasitoid can reduce abundance of the insect vectoring Xylella, but not its transmission ability of the pathogen. In the discussion, it should be clearly stated that data on the impact of egg parasitoids on pathogen spread are lacking.

Some references seem not properly cited, and some mistakes should be corrected (i.e. larva of P. spumarius instead of neanid/nymph).
Finally, it would be interesting to add information on sampling design. For example, why did the authors collect Cistus? Why only this plant, given that P. spumarius is a polyphagous species?

Reviewer 1 ·

Basic reporting

The article uses clear and unambiguous, professional English. The submission includes introduction and background enough to put the work fits into its broader field of knowledge. Relevant literature is appropriately referenced. The text has a proper structure, figures, tables and raw are offered for sharing, and are consistent and technically sounded to the content of the article.
The submission is self-contained with relevant results to hypotheses.

Experimental design

The proposal is primary research within the Aims and Scope of the journal, and the research question is well defined, relevant & meaningful.
The investigation must have been conducted to a high technical standard. The research must have been conducted in conformity with the prevailing ethical standards in the field.
Methods described with sufficient detail & information to replicate.

Validity of the findings

Impact and novelty are assessed.

not all the data on which the conclusions are based are made available in an acceptable discipline-specific repository. Some flaws in the data are discussed.
Conclusions are well stated, linked to original research question & limited to supporting results.
Speculation is identified as such.

Additional comments

Please consider the annotated .pdf or the Word file

Annotated reviews are not available for download in order to protect the identity of reviewers who chose to remain anonymous.

Reviewer 2 ·

Basic reporting

Well written and organized, interesting results. Please see attached pdf file with comments and suggested changes to improve the manuscript.

Experimental design

ok

Validity of the findings

ok

Additional comments

see attachment

Annotated reviews are not available for download in order to protect the identity of reviewers who chose to remain anonymous.

---

## Round 0.2 · Minor Revisions

The manuscript has now been substantially improved and needs only minor revisions prior to be published in PeerJ.

Authors deleted from the abstract the (over)statement that Ooctonus vulgatus “seems to be a promising biocontrol agent” for Philaenus spumarius, the vector of Xylella fastidiosa. The use of the term “potential” is more appropriate and coherent with data collected in the present research.

In the revised version, authors added significant details in materials and methods (for example, a clear definition of “parasitism rate”, why authors focused sampling of P. spumarius eggs on Cistus, etc.). Regarding the procedure for DNA extraction from O. vulgatus specimens, I would suggest authors to change the terms “destruction” and “dissection” with “crushing”. Indeed, DNA extraction “without destruction of the specimen” or from “whole insects, without dissection” could lead to misunderstanding, especially for readers not familiar with DNA extraction protocols from tiny insects (Mymarid species are about 1 mm long).
Authors deleted misleading sentences (e.g., the parasitoid cannot reduce the transmission ability of Xylella by P. spumarius but can reduce the potential for Xylella to spread) and improved both discussion and conclusions.

Please, change “larvae” of P. spumarius also in captions of legends: juvenile stages of the spittlebug are neanids and nymphs.

Reviewer 1 ·

Basic reporting

English is clear, unambiguous, and technically correct. Courtesy and expression is respected. References are sufficient for the context.

Introduction and background demonstrate the work fits into the broader field of knowledge, prior literature is appropriately referenced. There is enough structure, figures, and tables.

The article is structured in standard sections.

Figures are not abundant but enough relevant to the content of the article.

Any raw picture is available, the study is self-contained with appropriate results to hypotheses.

The coherent bodies of work is appropriately subdivided.

Experimental design

This is one original primary research within the Aims and Scope of the journal.
The research question is well defined, relevant & meaningful. It is stated how research fills an identified knowledge gap.

The submission clearly defines the relevant and meaningful research question. The knowledge gaps are identified and investigated, the study contributes to filling that gap.
Rigorous investigation performed to a high technical & ethical standard.

The investigation has a rigorous and high technical standard, given in conformity with the prevailing ethical standards in the field. The methods are described with sufficient detail & information to replicate to be reproducible by another investigator.

Validity of the findings

The submission defines a field of interest and encourages to explore the topic by a rational approach.

The data on which the conclusions are based are available in an acceptable discipline-specific repository. The data should be robust, statistically sound, and controlled.
Conclusions are well stated, linked to original research question & limited to supporting results.

The conclusions are appropriately stated, connected to the original question investigated, and limited to those supported by the results.

Speculations are identified as such.

Additional comments

please consider the comments added into the U/L .pdf that comprehends the submission, the rebuttal letter, and the AppendixS1

Annotated reviews are not available for download in order to protect the identity of reviewers who chose to remain anonymous.

Reviewer 2 ·

Basic reporting

ok

Experimental design

ok

Validity of the findings

ok

Additional comments

Please find below my comments on the manuscript entitled “Ooctonus vulgatus (Hymenoptera, Mymaridae), a potential biocontrol agent to reduce populations of Philaenus spumarius (Hemiptera, Aphrophoridae) the main vector of Xylella fastidiosa in Europe”. The manuscript is well written and organized. I found no flaws that make me doubt of the author’s interpretation of results. Please find below some minor editorial changes to improve the manuscript.

L30: Replace “spectrum” by “range”
L33: Change to “This report serves as a …”
L42-43: Change to “wine and grape industries.”
L61-63: Check journal guidelines for citations with more than two authors. Should these be cited as “Halkka et al. 1975” and so on?
L64: Replace “recently” by “previously”
L65: Add “Severin 1950” here and in the references cited. Severin, H.H.P. 1950. Spittle-insect vectors of Pierce’s disease virus. Hilgardia 19: 357-382.
L79: Delete “Carabidae” as no other families were reported.
L82: Provide authorship and Order for Zelus renardii
L92: Change to “…because it renders the host…”
L103: Replace “efficiency” by “efficacy”
L108-109: Add parenthesis to “i.e. geographical coordinates…O. vulgatus”
L118: Change “Cistus” to “C.”
L138: Delete “50 fps”
L219: Replace “came out” by “emerged”
L230: Replace “discriminated” by “distinguished”
L264: Place countries in parenthesis
L270: What is 197? Please explain or delete.
L286: Replace “what are the drivers” by “what is the cause(s)”.
L292, 298, and 300: Provide Order and Family for Gonipterus scutellatus, Edwardsiana froggatti, and Homalodisca vitripennis, respectively.
L299: I suggest changing “Gonatocerus” to “Cosmocomoidea”. See Zootaxa 3967 (1): 001–184 for the “World reclassification of the Gonatocerus group of genera (Hymenoptera: Mymaridae)” by John Huber”.
L304: Replace “addressed” by “determined”
L554: Replace “larvae” by “nymphs”
L555: The pie charts have only P. spumarius. It does not show the dark and light grey portions for adults and nymphs.

---

## Round 0.3 · accepted · Accept

In my opinion, the manuscript is now ready to be published. I congratulate the authors for their effort.